# How Does the Volatility of Volatility Depend on Volatility?

**Sigurd Emil Rømer** and **Rolf Poulsen** *

Department of Mathematical Sciences, University of Copenhagen, 2100 København Ø, Denmark;
ser@math.ku.dk

* Correspondence: rolf@math.ku.dk

**Abstract:** We investigate the state dependence of the variance of the instantaneous variance of the S&P 500 index empirically. Time-series analysis of realized variance over a 20-year period shows strong evidence of an elasticity of variance of the variance parameter close to that of a log-normal model, albeit with an empirical autocorrelation function that one-factor diffusion models fail to capture at horizons above a few weeks. When studying option market behavior (in-sample pricing as well as out-of-sample pricing and hedging over the period 2004–2019), messages are mixed, but systematic, model-wise. The log-normal but drift-free SABR (stochastic-alpha-beta-rho) model performs best for short-term options (times-to-expiry of three months and below), the Heston model—in which variance is stationary but not log-normal—is superior for long-term options, and a mixture of the two models does not lead to improvements.

**Keywords:** stochastic volatility; elasticity of variance of variance; Heston; SABR

## 1. Introduction

The aim of this paper is to investigate the elasticity of variance of variance of the S&P 500 index. As that concept is not part of the standard financial nomenclature[1], let us introduce the general set-up. We consider a probability space $(\Omega, \mathcal{F}, (\mathcal{F}_t)_{t \geq 0}, \mathbb{P})$ whose filtration is generated by two independent Brownian motions $W_{1,t}$ and $W_{\perp,t}$. Letting $S_t$ denote the value of the S&P 500 index at time $t$, we model dynamics under the real world probability measure $\mathbb{P}$ as:

$$dS_t = \mu_t^s S_t dt + \sqrt{V_t} S_t dW_{1,t} \tag{1}$$

with

$$dV_t = \mu_t^v dt + \eta V_t^{\lambda_v} dW_{2,t} \tag{2}$$

and where we have defined the Brownian motion $W_{2,t} = \rho W_{1,t} + \sqrt{1-\rho^2} W_{\perp,t}$ for a $\rho \in [-1, 1]$, meaning that $\langle dW_{1,t}, dW_{2,t} \rangle = \rho dt$. We assume that $\mu_t^s$ and $\mu_t^v$ are general (but suitable) adapted processes. We refer to $V_t$ as the (instantaneous) variance. Our main focus will be the elasticity parameter[2] $\lambda_v$, which controls the level dependence of variance-of-variance with respect to the instantaneous variance $V_t$ itself. We will see that determining $\lambda_v$ is not merely an academic exercise. Empirically the parameter choice is important for model performance, including the effective hedging of options.

---

[1]  A Google search on "elasticity of variance of variance" in late March 2020 gave only seven hits.

[2]  Mathematically, the elasticity of a function $f$ is $xf'(x)/f(x)$. The *variance* of variance $V_t$ is $\eta^2 V_t^{2\lambda_v}$, and thus pedants can rightly claim that $2\lambda_v$ is the elasticity of variance of variance.

There is an abundance of both theoretical and empirical work on stochastic volatility in financial markets. However, as succinctly formulated by Felpel et al. (2020), "Often it is observed that a specific stochastic volatility model is chosen not for particular dynamical features, but instead for convenience and ease of implementation". In this short paper we aim for an empirical cross-examination of models, methods, and markets. It means that either of the separate analyses may be described—or dismissed—as "quite partial", but we believe that the sum of their parts brings to the fore some insights that were not hitherto available. In Section 2 we estimate the elasticity from a time series of realized variance, thus extending the analysis in Tegnèr and Poulsen (2018), where only pre-specified values of elasticity were considered. We end the section with an investigation of features that are not captured by one-factor diffusion-type stochastic volatility models, or long-memory or long-range dependence in particular. In Section 3 we turn our attention to option prices to investigate the two—by far—most widely used models, the square-root model from Heston (1993) and the log-normal SABR (stochastic-alpha-beta-rho) model from Hagan et al. (2002), as well a simple hybrid of the two models. We study in-sample calibration issues along the lines of Guillaume and Schoutens (2010) as well as predition and hedge performance, adding to the analyses in Rebonato (2020) and Poulsen (2020).

## 2. Evidence from Realized Variance

In this section we present four different methods for estimating $\lambda_v$ from time series data, investigate their biases in a simulation experiment, and finally show the estimates on publicly available empirical data—the realized variance obtained from the Oxford-Man Institute's "realized library".[3] In Tegnèr and Poulsen (2018) it is demonstrated that it is possible to discriminate between common diffusion-type stochastic volatility models when measuring realized variance from a 5-minute observation frequency, so we chose that data (denoted "rv5" in the files) among the many time series in the Oxford-Man data.[4] Our dataset thus consists of daily observations between the 3rd of January 2000 and the 3rd of September 2019. After filtering out non-positive variance estimates and removing the day of the Flash Crash (6th of May 2010) we were left with a total of 4935 observations. We should also remark that since we focus on the $\lambda_v$-parameter, the estimation methods will all solely focus on estimating the variance (or volatility) process. That is, we leave out a joint estimation also including the index price.

Let us now start by assuming a mean-reverting variance model of the form

$$dV_t = \kappa(\theta - V_t)dt + \eta V_t^{\lambda_v}dW_{2,t} \tag{3}$$

with $\kappa, \theta > 0$ being additional parameters. To estimate the parameters we can apply an Euler discretisation and then use maximum likelihood. Let $t_0 < t_1 < ... < t_n$ with $n = 4934$ denoting the observation time points. We will, for simplicity, assume the time points are equidistant with step sizes of $\Delta t = 1$ trading day with 252 trading days assumed per year. We then approximate

$$\Delta V_{t_i} = \kappa(\theta - V_{t_i})\Delta t + \eta V_{t_i}^{\lambda_v}\Delta W_{2,t_i} \tag{4}$$

for $i = 0, 1..., n-1$, where we write $\Delta X_{t_i} = X_{t_{i+1}} - X_{t_i}$ for a general process $X$. Under the approximative model (4), the joint density of $(V_{t_1}, ..., V_{t_n})$ is a product of conditional densities, which are all Gaussian. To obtain maximum approximate likelihood estimates we plug in the observed time series and numerically maximize the log of this joint density, the log-likelihood function, over the model

---

[3]  https://realized.oxford-man.ox.ac.uk/

[4]  The use of 5-minute observations to measure instantaneous variance is common and we think it is a reasonable compromise, but we should stress that it does not tell the whole story about what goes on at higher frequencies; see for instance Christensen et al. (2019).

parameters[5] and use the Hessian matrix of the log-likelihood at the maximum to give us standard errors. Another model specification is to use a mean-reverting model for the *volatility process* $\sigma_t := \sqrt{V_t}$ instead,

$$d\sigma_t = \kappa(\theta - \sigma_t)dt + \eta\sigma_t^{\lambda_\sigma}dW_{2,t}. \tag{5}$$

Ito's formula applied to the squared solution of (5) reveals that while the mean-reverting volatility and variance models are not equivalent as the functional forms of the drifts are different, we do have

$$dV_t = ...dt + 2\eta V_t^{(\lambda_\sigma+1)/2}dW_{2,t} \tag{6}$$

so that $\lambda_v = (\lambda_\sigma + 1)/2$ and the maximum approximate likelihood technique applied to the volatility process gives us an alternative way to estimate the elasticity.[6] The final two estimators we will consider are based on the concept of quadratic variation. Following proposition 4.21 of Le Gall (2016) the quadratic variation of a continuous semimartingale $X$ over the interval $[0, t]$, here denoted $\langle X, X \rangle_t$, can be characterised as follows: Let $0 = t_0^n < t_1^n < ... < t_{p(n)}^n = t$ with $p(n) \in \mathbb{N}$ for $n = 1, 2, ...$ be an increasing sequence of partitions of $[0, t]$ satisfying $\sup_{1 \leq i \leq p(n)} |t_i^n - t_{i-1}^n| \to 0$ as $n \to \infty$. Then

$$\langle X, X \rangle_t = \lim_{n \to \infty} \sum_{i=1}^{p(n)} \left( X_{t_i^n} - X_{t_{i-1}^n} \right)^2 \tag{7}$$

with convergence in probability. For the $V_t$ in the general model (2) the quadratic variation is:

$$\langle V, V \rangle_t = \int_0^t \eta^2 V_s^{2\lambda_v} ds. \tag{8}$$

Combining (7) and (8) leads to the approximation $(V_{t_i} - V_{t_{i-1}})^2 \approx \eta^2 V_{t_{i-1}}^{2\lambda_v} \Delta t$, or in logarithm terms

$$\log((V_{t_i} - V_{t_{i-1}})^2) \approx 2\log(\eta) + 2\lambda_v \log(V_{t_{i-1}}) + \log(\Delta t). \tag{9}$$

This means we can estimate $\lambda_v$ and $\eta$ from a linear regression between the samples $X_i := \log(V_{t_{i-1}})$ and $Y_i := \log((V_{t_i} - V_{t_{i-1}})^2)$. Applying the same reasoning to the squared increments of the volatility process,

$$\log((\sigma_{t_i} - \sigma_{t_{i-1}})^2) \approx 2\log(\eta) + 2\lambda_\sigma \log(\sigma_{t_{i-1}}) + \log(\Delta t), \tag{10}$$

gives way to estimate $\lambda_\sigma$ and $\eta$ by a linear regression.

The four estimators presented all suffer from several sources of bias: discretization bias, small sample bias, and a possible error from model misspecification; if the variance has linear drift, then the volatility does not, and vice versa. Hence to validate the estimators we conducted a simulation experiment, whose results, shown in Table 1, we shall now briefly describe. For the experiment we assumed either the mean-reverting volatility or the mean-reverting variance model to be the true model, and simulated 1000 sample paths with 2520 steps per year across a total of 4935 days, thus mimicking the empirical set-up. Using only daily observations, we then applied each of the four methods to estimate $\lambda_v$ and $\lambda_\sigma$ and averaged across the paths. We performed this entire experiment for different choices of $\lambda_v \in \left[\frac{1}{2}, 1\right]$ and $\lambda_\sigma \in \left[\frac{1}{2}, 1\right]$. For a realistic set-up, the remaining parameters (except for $\theta$ under (3)) were chosen by running the relevant likelihood estimation on the Oxford-Man dataset

---

[5] We exclude here the $\theta$-parameter by instead fixing it at the time series mean of $V_t$. We find this produces a more reasonable estimation.

[6] In case $V_t$ from Equation (3) can hit zero, applying the $\sqrt{x}$-function to get back an elasticity for $\sigma_t$ is problematic. However, all our estimators suggest $\lambda_v$ is above $\frac{1}{2}$, in which case zero (see for instance Andersen and Piterbarg (2007)) is unattainable, so we will use the relationship $\lambda_v = (\lambda_\sigma + 1)/2$ without worry.

with $\lambda_v$ or $\lambda_\sigma$ held fixed at the true value. The experiment shows that both regression estimators are fairly robust under the two different models as well as across different elasticities. The same is true when using maximum approximate likelihood on Equation (5). For these three estimators, the bias when estimating $\lambda_v$ is mostly less than 0.05. Maximum approximate likelihood estimation on Equation (3) performed much worse, with a consistent downwards bias of 0.1–0.2 depending on the true modeling assumptions and the elasticity used.

In Table 2 we show estimates on the actual dataset using each of the four methods. We include the $R^2$ statistic for the regression methods and show standard errors in parenthesis. There is a high correspondence between the elasticities estimated under all methods—except using likelihood on Equation (3), which we just saw to be problematic. We found $\lambda_v$-values in the range 0.91–0.97. In this range, the bias is still mostly less than 0.05 and the estimated standard errors are less than 0.02, so we are quite confident that the elasticity is in that range. The estimates in Table 2 also show that instantaneous variance is characterized by a combination of strong mean-reversion and high volatility of volatility (both $\kappa$ and $\eta$ are high compared to many other sources), which—if one were to think briefly beyond the realm of diffusion models—could point towards so-called rough volatility models as suggested by Gatheral et al. (2018).[7]

We end this section by looking into some of the problems with modeling volatility by a one-dimensional diffusion process. First, the long-memory properties of volatility that have been widely documented; an example is the paper Bennedsen et al. (2016). To have a precise discussion let us introduce some notation: Consider a covariance-stationary process $X$ and pick an arbitrary time point $t$. We then define the autocorrelation function of the $X$-process at lag $h$ as:

$$\text{ACF}(h) = \frac{\text{Cov}(X_t, X_{t+h})}{\text{Var}(X_t)}. \tag{11}$$

Several slightly different definitions of long-memory can be found in the literature. As an example, in Bennedsen et al. (2016) the autocorrelation function of the log-volatility is assumed to decay as $\text{ACF}(h) \sim c|h|^{-\beta}$ for $|h| \to \infty$, where $c, \beta > 0$ are constants. The volatility process is then said to have long-memory exactly if $\int_0^\infty |\text{ACF}(h)| dh = \infty$, which happens when $\beta \in (0,1)$. In our context, with daily observations, a reasonable test for long-memory is therefore to see if the sums

$$\sum_{i=0}^k \widehat{\text{ACF}}\left(i/252\right), \text{ for } k = 0,1,2... \tag{12}$$

converge, where $\widehat{\text{ACF}}(h)$ denotes an empirical estimate of (11).

In Figure 1, we show the sums in (12) for lags up to around 1 year. Specifically, the blue and orange lines show the sums (12) for the realized variance and volatility, respectively, both obtained from the Oxford-Man dataset. The lines indicate that there is detectable autocorrelation even at horizons of up to one year (although it is most pronounced looking at realized *volatility*). This can then be compared to what our diffusion model specifications from (3) and (5) are able to produce. To this end, we conducted a simulation experiment, where we simulated each model in the same way as done for the bias-experiment shown in Table 1, though this time using the estimated parameters from Table 2. For each such simulated path we then computed the empirical autocorrelation function $\widehat{\text{ACF}}(h)$ at various lags $h$. Repeating the simulations 10,000 times, averaging the autocorrelations and computing the running sums in Equation (12), we get the yellow line (simulation of Equation (3)) and purple line (simulation of Equation (5)). Here it is clear that while the autocorrelations closely match the empirical ones for the first few lags, the fit is very bad at longer horizons—where in fact "longer" does not have to mean more than a few weeks.

---

[7] A very useful source for the rapidly expanding field of rough volatility is https://sites.google.com/site/roughvol/home.

**Table 1.** The accuracy when estimating the elasticity parameters $\lambda_v$ and $\lambda_\sigma$ under various modeling assumptions (mean-reverting variance or volatility) and various estimation techniques (maximum approximate likelihood estimation and regressions). The experiment was conducted as follows: assuming either Model (3) or Model (5), we first simulated $N = 1000$ independent paths across 4935 days with 10 simulation steps per day. Here we used the locally log-normal scheme from Andersen and Brotherton-Ratcliffe (2005) as presented in Lord et al. (2010). Using only the 4935 daily values we then path-by-path estimated the $\lambda_v$ and $\lambda_\sigma$ parameters using each of the four estimation techniques. This gave us 1000 estimates for each, which we write as $\{\hat{\lambda}_{v,i}\}_{i=1}^N$ and $\{\hat{\lambda}_{\sigma,i}\}_{i=1}^N$, respectively. The goal is then to estimate the means $E(\hat{\lambda}_{v,1})$ and $E(\hat{\lambda}_{\sigma,1})$ for each estimation technique. These values should then be compared to the true values used to simulate the paths (columns 4 and 9 below) to see how biased each method is. The means can naturally be estimated by the averages $\bar{\lambda}_\sigma = \frac{1}{N}\sum_{i=1}^N \hat{\lambda}_{\sigma,i}$ (columns 5–8) and $\bar{\lambda}_\sigma = \frac{1}{N}\sum_{i=1}^N \hat{\lambda}_{\sigma,i}$ (columns 10–13). To get an idea of how reliably $\bar{\lambda}_v$ and $\bar{\lambda}_\sigma$ estimate the means of each method, we can also compute standard errors as $\hat{v}_v/\sqrt{N}$ and $\hat{v}_\sigma/\sqrt{N}$, where we have defined $\hat{v}_v = \sqrt{\frac{1}{N-1}\sum_{i=1}^N \left(\hat{\lambda}_{v,i} - \bar{\lambda}_v\right)^2}$ and $\hat{v}_\sigma = \sqrt{\frac{1}{N-1}\sum_{i=1}^N \left(\hat{\lambda}_{\sigma,i} - \bar{\lambda}_\sigma\right)^2}$. While we do not show the standard errors, we can report that they were all less than 0.01.

True model: $dV_t = \kappa(\theta - V_t)dt + \eta V_t^{\lambda_v} dW_{2,t}$

| $\kappa$ True Value | $\theta$ True Value | $\eta$ True Value | $\lambda_v$ True Value | $\bar{\lambda}_v$ ML MR var. | $\bar{\lambda}_v$ ML MR vol. | $\bar{\lambda}_v$ Regression on var. | $\bar{\lambda}_v$ Regression on vol. | $\lambda_\sigma$ True value | $\bar{\lambda}_\sigma$ ML MR var. | $\bar{\lambda}_\sigma$ ML MR vol. | $\bar{\lambda}_\sigma$ Regression on var. | $\bar{\lambda}_\sigma$ Regression on vol. |
|---|---|---|---|---|---|---|---|---|---|---|---|---|
| 25.56 | $0.16^2$ | 2.31 | **0.50** | 0.35 | 0.63 | 0.59 | 0.68 | **0.00** | $-0.31$ | 0.25 | 0.19 | 0.36 |
| 22.34 | $0.16^2$ | 3.27 | **0.60** | 0.44 | 0.68 | 0.65 | 0.72 | **0.20** | $-0.12$ | 0.36 | 0.29 | 0.44 |
| 20.25 | $0.16^2$ | 4.82 | **0.70** | 0.54 | 0.75 | 0.72 | 0.78 | **0.40** | 0.08 | 0.49 | 0.43 | 0.57 |
| 18.64 | $0.16^2$ | 7.33 | **0.80** | 0.65 | 0.82 | 0.79 | 0.86 | **0.60** | 0.29 | 0.64 | 0.58 | 0.71 |
| 17.25 | $0.16^2$ | 11.49 | **0.90** | 0.75 | 0.89 | 0.87 | 0.93 | **0.80** | 0.50 | 0.79 | 0.73 | 0.87 |
| 16.01 | $0.16^2$ | 18.52 | **1.00** | 0.86 | 0.96 | 0.93 | 1.01 | **1.00** | 0.71 | 0.93 | 0.86 | 1.02 |

True model: $d\sigma_t = \kappa(\theta - \sigma_t)dt + \eta \sigma_t^{\lambda_\sigma} dW_{2,t}$

| $\kappa$ True Value | $\theta$ True Value | $\eta$ True Value | $\lambda_v$ True Value | $\bar{\lambda}_v$ ML MR var. | $\bar{\lambda}_v$ ML MR vol. | $\bar{\lambda}_v$ Regression on var. | $\bar{\lambda}_v$ Regression on vol. | $\lambda_\sigma$ True Value | $\bar{\lambda}_\sigma$ ML MR var. | $\bar{\lambda}_\sigma$ ML MR vol. | $\bar{\lambda}_\sigma$ Regression on var. | $\bar{\lambda}_\sigma$ Regression on vol. |
|---|---|---|---|---|---|---|---|---|---|---|---|---|
| 40.29 | 0.13 | 2.05 | **0.75** | 0.55 | 0.72 | 0.69 | 0.74 | **0.50** | 0.09 | 0.44 | 0.37 | 0.48 |
| 38.98 | 0.13 | 2.50 | **0.80** | 0.62 | 0.77 | 0.74 | 0.79 | **0.60** | 0.23 | 0.53 | 0.48 | 0.58 |
| 37.69 | 0.14 | 3.07 | **0.85** | 0.69 | 0.81 | 0.79 | 0.84 | **0.70** | 0.38 | 0.63 | 0.59 | 0.68 |
| 36.39 | 0.14 | 3.79 | **0.90** | 0.76 | 0.86 | 0.85 | 0.89 | **0.80** | 0.51 | 0.73 | 0.69 | 0.77 |
| 35.09 | 0.14 | 4.72 | **0.95** | 0.82 | 0.91 | 0.90 | 0.94 | **0.90** | 0.64 | 0.82 | 0.79 | 0.87 |
| 33.76 | 0.14 | 5.92 | **1.00** | 0.90 | 0.96 | 0.95 | 0.98 | **1.00** | 0.79 | 0.93 | 0.89 | 0.97 |

**Column descriptions:** * ML MR var. = (average) maximum approximate likelihood estimate assuming variance is mean reverting; * ML MR vol. = (average) maximum approximate likelihood estimate assuming volatility is mean reverting; * Regression on var. = (average) regression estimate when using the quadratic variation of variance; * Regression on vol. = (average) regression estimate when using the quadratic variation of volatility.

**Table 2.** Estimated parameters under each of the four estimation methods discussed. Standard errors are shown in parenthesis where applicable.

| Method | $\kappa$ | $\theta$ | $\eta$ | $\lambda_v$ | $\lambda_\sigma$ | $R^2$ |
|---|---|---|---|---|---|---|
| Max. approx. likelihood on Equation (3) | 18.62 (1.07) | $0.16^2$ | 7.37 (0.27) | 0.80 (0.01) | 0.60 (0.02) | |
| Max. approx. likelihood on Equation (5) | 36.01 (2.65) | 0.14 (0.01) | 4.04 (0.16) | 0.91 (0.01) | 0.83 (0.02) | |
| Regression on Equation (9) | | | 3.92 (0.27) | 0.93 (0.02) | 0.85 (0.03) | 0.42 |
| Regression on Equation (10) | | | 2.35 (0.16) | 0.97 (0.01) | 0.94 (0.03) | 0.17 |

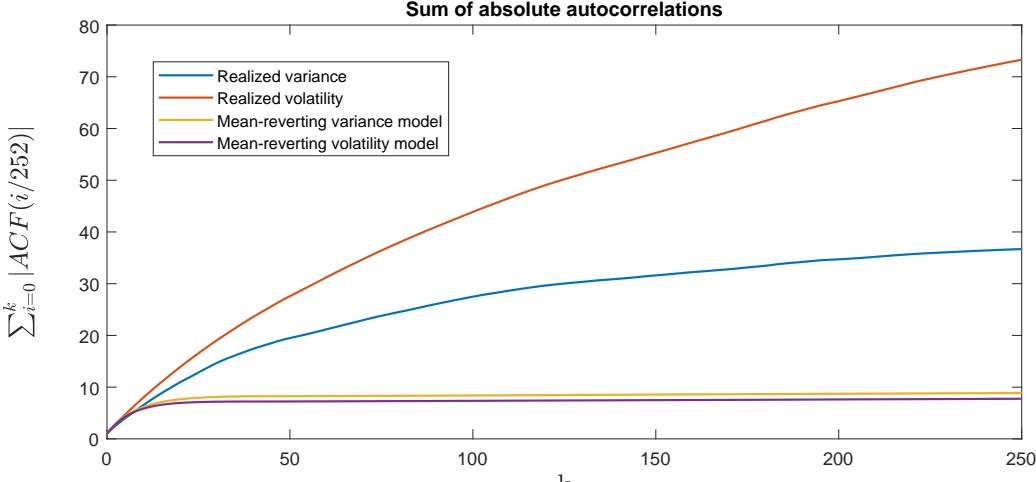

**Figure 1.** Cumulative sums of the empirical (absolute-value) autocorrelations. Blue and orange lines show the sums for the realized variance and volatility and that using the Oxford-Man dataset, respectively. The yellow and purple lines show the sums for the model (3) and (5), respectively. These values are computed in a simulation experiment using the parameters from Table 2. For these curves, the standard errors of each autocorrelation value were all less than 0.01.

To interpret the figure we once again cite Bennedsen et al. (2016), where it was assumed that the autocorrelation function of log-volatility behaves as $1 - \mathrm{ACF}(h) \sim c|h|^{2\alpha+1}$ when $|h| \to 0$ and where $c > 0$ and $\alpha \in (-1/2, \infty)$ are constants. Volatility is then said to be rough when $\alpha \in (-1/2, 0)$. That is, roughness is exactly related to how the autocorrelation function behaves for very short lags. With this knowledge and together with our high $\kappa$ and $\eta$ estimates we interpret the good fits for short lags in Figure 1 as evidence that our one-factor diffusion models have attempted to mimic roughness. The model-structure then does not allow us to simultaneously capture the memory in the process at medium and long lags.

An extension that would allow for both roughness and long-memory would be to model $V_t$ (alternatively $\sigma_t$) as a mean-reverting stochastic Volterra equation (SVE).[8] Specifically, one could model

$$V_t = V_0 + \int_0^t K(t-s)\kappa\,(\theta - V_s)\,ds + \int_0^t K(t-s)\eta V_s^{\lambda_v}dW_{2,s}, \tag{13}$$

where $K$ is a kernel function that if chosen appropriately and exactly would allow $V_t$ to display both of these properties. Although an estimation of the more general Equation (13) would certainly be a worthwhile pursuit, we leave it as an open hypothesis whether or not this will change the $\lambda_v$-estimates.

---

[8]　The reader can consult Abi Jaber et al. (2019) for some mathematical theory on stochastic Volterra equations.

Two other examples—among several—of models that can capture long-memory are Zumbach (2004) and Borland and Bouchaud (2011).

Another objection to our models is the lack of jumps in the asset price. However, as shown in Christensen et al. (2014) using high-frequency data on numerous different financial assets, the jump proportion of the total variation of the asset price is generally small compared to the volatility component—at least when market micro-structure effects are properly accounted for. It is therefore very possible that adding jumps to the asset price will only slightly change the $\lambda_v$-estimates. On the other hand, one could also consider jumps in the volatility process. Then again, with the current evidence in favor of rough volatility one could hypothesize that the most variation in the true volatility process can also be explained by continuous but rough (i.e., explosive) movements and not jumps. As with the long-memory objections, we leave a proper analysis for future research.

## 3. Evidence from Option Prices

We now turn our attention to options, more specifically to the market for European call and put options on the S&P 500 index. In an ideal world, one would specify flexible parametric structures for the drift function in Equation (2) and for the market price of volatility risk, solve the associated pricing PDE, and then choose the parameter-values that minimize the distance—suitably measured—between market data and model prices. However, genuinely efficient methods for option price calculations exist only for a few quite specific models. We shall restrict our interest to the two most common models, the square-root model from Heston (1993) that sparked a revolution in affine models and transform methods, and the SABR (stochastic-alpha-beta-rho) model from Hagan et al. (2002) that made stochastic volatility a household object in banks.

We used the *End-of-Day Options Quote Data* obtained from https://datashop.cboe.com. This dataset contains bid and ask quotes on SPX European options at 15:45 Eastern Time (ET) and again at the close of the market. We used the 15:45 quotes for liquidity reasons and applied a number of filters to (a) clean the data and (b) compute the zero coupon bond and dividend yields implied by the put-call parity (and used in the subsequent analysis). Finally, we used smoothing and interpolation techniques on the mid quotes to obtain prices on a continuous set of strikes on each of the fixed set of expiries 1, 3, 6, 12, 18, and 30 months.[9] The cleaned dataset contains observations on 3783 trading days between the 3rd of May 2004 and the 15th of May 2019.

We will now briefly define the pricing models we use in our experiments. The models are therefore stated under an equivalent risk-neutral probability measure $\mathbb{Q}$. Note that Girsanov's theorem tells us that the elasticity parameter is unaffected by equivalent measure changes. Thus, there is no a priori conflict between $\lambda_v$-estimates obtained from time series observations of $V_t$ ("under $\mathbb{P}$") from the previous section and risk-neutral values affecting option prices that we shall be looking at in this section. For simplicity we will abuse notation and write $W_{1,t}$ and $W_{2,t}$ to also denote two $\mathbb{Q}$-Brownian motions s.t. $\langle dW_{1,t}, dW_{2,t} \rangle = \rho dt$ for a $\rho \in [-1, 1]$. Denoting the interest rate by $r$ and the dividend yield by $q$ we then have

$$dS_t = (r - q)S_t dt + \sqrt{V_t} S_t dW_{1,t}. \tag{14}$$

Under the Heston model of Heston (1993), the variance process $(V_t)_{t \geq 0}$ is then modeled as a square-root diffusion of the form

$$dV_t = \kappa(\bar{v} - V_t)dt + \eta\sqrt{V_t}dW_{2,t} \tag{15}$$

with $\kappa, \bar{v}, \eta, V_0 > 0$. The model thus has elasticity $\lambda_v = \frac{1}{2}$, which (Feller condition issues notwithstanding) is equivalent to $\lambda_\sigma = 0$. We compute option prices using numerical integration with

---

9    We used the arbitrage free smoothing spline from Fengler (2009) on expiry-slice.

the techniques from Lord and Kahl (2006). The SABR model from Hagan et al. (2002) (here with their $\beta = 1$) instead assumes

$$d\sigma_t = \alpha \sigma_t dW_{2,t} \tag{16}$$

with $\alpha, \sigma_0 > 0$. To compute option prices we use here the approximation formula also found in Hagan et al. (2002). In elasticity-terms, the log-normal SABR model has $\lambda_v = \lambda_\sigma = 1$. Finally, we consider a third hybrid or mixture model, where we assume that the price of any vanilla European option is the average of the prices under both the Heston and SABR models. Specifying such a mixture model is exactly equivalent to specifying the marginal distributions of $S_t$ for all $t > 0$ but nothing more. The mixture model allows us to incorporate a marginal distribution for $S_t$ somewhere between Heston and SABR. The model is *underspecified*, as it does not contain information on the dynamical structure of the asset price $S_t$; see the paper Piterbarg (2003) on mixture models and their pitfalls. However, because vanilla option prices under the mixture model are convex combinations of prices from the arbitrage-free models Heston and SABR, the mixture model is free from static arbitrages; it could thus at the very least be supported by a local volatility model for $S$.[10]

As explained, our choice of pricing models has been restricted to those that are tractable. As a downside, each model has its own problems in terms of matching the empirical stylized facts of volatility. Some of these problems are related to the topic of this paper, i.e., what should the elasticity of variance-of-variance be. Specifically, in Section 1 we estimated $\lambda_v \approx 1$ while Heston has $\lambda_v = 1/2$. Likewise, in Tegnèr and Poulsen (2018) it is shown that while a log-normal model fits the marginal distribution of volatility well, Heston does not. Given our topic, having differences in $\lambda_v$ is of course exactly as desired. In an ideal controlled experiment this would be the only difference between the models. This is not quite true. The Heston model has mean-reversion and SABR does not. Furthermore, any inability to match empirical facts, even if shared by both models, could make other parts of the models and the results move in unexpected and hard-to-explain ways. As an example, both models suffer from the inability to match the entire term structure of autocorrelations (as shown in Section 1). With all of this taken together, it would be naive to think that we can reasonably fit and/or accurately model multiple expiries at once. In an attempt to mend some of these problems we therefore chose to model each expiration-slice with a separate model.[11] The hope is that this will help the models capture the temporal properties that are the most important to modeling the particular expiration in question. Ideally this would also mean that the models approximately only differ by their elasticity. The reality is of course not so simple. We will make our best attempt at analyzing the results despite these a-priori objections.

### 3.1. Calibration and In-Sample Model Performance

In this subsection we perform a calibration experiment, where we calibrate the models separately on each expiry. We perform the calibration on a given expiry by minimizing the mean absolute error in Black–Scholes implied volatilities across a number of observed contracts, $n$. Let $\widehat{\sigma}_i^{obs}$ denote the implied volatility of the $i$'th contract as observed on the market, and let $\widehat{\sigma}_i^{model}$ denote the corresponding implied volatility for the model in question and that for a particular choice of parameters. We thus calibrate each model by minimizing

$$\frac{1}{n} \sum_{i=1}^{n} \left| \widehat{\sigma}_i^{obs} - \widehat{\sigma}_i^{model} \right| \tag{17}$$

---

[10] Lewis (2019) presents a fully specified mean-reverting, lognormal-ish volatility model that allows for explicit call option prices. However, in the author's own words, "The bad news is that the integrand requires the infinite sum at (37). I truncate that sum, and make some use of Mathematica's `Parallelize`. This case is really computationally tedious: results can take a half-hour or more." Hence the model is not suitable for our analysis.

[11] The $r$ and $q$ values of Equation (14) will therefore be the interest rate yield and dividend yield for the particular expiration in question.

with respect to the model parameters. On each day and for each expiry we aim to calibrate to 7 options: those with Black–Scholes call Deltas 0.05, 0.2, 0.3, 0.5, 0.7, 0.8, and 0.95.[12] When calibrating the Heston model to a single expiry-slice it is difficult to separate the speed of mean-reversion parameter $\kappa$ from the volatility-of-volatility parameter $\eta$, as well as separating the current instantaneous variance $V(t)$ from its long-term mean $\bar{v}$; see also Guillaume and Schoutens (2010). Hence we use the following two-step procedure: first, optimize the fit to all implied volatilities over $(\kappa, \eta, \rho, V(t), \bar{v})$; second, keep the $\kappa$-estimate from the first step fixed and then optimize each expiry-slice separately and under the constraint $V(t) = \bar{v}$.

In the left-hand side of Table 3 we show average calibration errors in basis points (bps) as well as their standard deviations (shown in parentheses). The main observation is that on average, SABR calibrates better for short expiries (less than 6 months) and Heston calibrates better for long expiries (more than one year). Specifically, the average error for the 1 month expiry is 28.7 bps for Heston and 11.2 bps for SABR, whereas for the 2.5 years' expiry, the average error is 14.2 bps for Heston and 23.6 bps for SABR.[13] To understand this result, remember (see Romano and Touzi (1997)) that call option prices are determined by the (risk-neutral) distribution of instantaneous variance integrated over the life-time of the option ($\tau$),

$$\frac{1}{\tau} \int_t^{t+\tau} \sigma_u^2 du. \tag{18}$$

For short times to expiry (small $\tau$), the dominant feature in determining this distribution is the $dW$-term in the dynamics of instantaneous volatility, while for longer times to expiry temporal dependence (such as mean-reversion) becomes more important. The empirical analysis in Section 2 shows that the SABR model (with $\lambda_v = 1$) captures the functional form of the variance of instantaneous variance better than the Heston model (that has $\lambda_v = \frac{1}{2}$). On the other hand, that analysis also shows that instantaneous variance has a quite significant mean-reversion, which is something that the Heston model captures but the SABR model does not. As Figure 2 shows, there is, however, a subtlety to this; the risk-neutral ($\mathbb{Q}$) speed of mean-reversion is considerably lower (2–4 typically) than the real-world ($\mathbb{P}$) estimates (18.62 from Table 2 is directly comparable).[14] It is widely documented in the literature that the short-expiry at-the-money implied volatility is typically higher than realized volatility, i.e., long-term levels are different between $\mathbb{P}$ and $\mathbb{Q}$ ($\bar{v} > \theta$), which can be explained by investor risk-aversion as a stochastic volatility model is, in option pricing terms, incomplete. The effect of investor preferences on the speed of mean-reversion is however less well documented. With that mean-reversion vs. elasticity reasoning in mind, a natural conjecture would be that a mixture model, a convex combination of SABR and Heston, would outperform both these models; that there would be a benefit from model diversification. However, such an effect is far from evident in the data. With the 6-month horizon as the only exception, the errors from the mixture model fall between those of SABR and Heston—often right in the middle.

---

[12]  To be precise: on a given day and for a given expiry we first attempt to interpolate the values 0.3, 0.5, and 0.7 (if that is not possible we discard that expiry on that day). Next we attempt to choose Black–Scholes deltas closest to the remaining shown values on each of the intervals [0.05,0.15], [0.15,0.30], [0.7,0.85], and [0.85,0.95]. Depending on the available range of strikes we thus in practice (on a smaller number of days) calibrate to fewer than seven quotes.

[13]  For comparison, in our dataset (and before interpolation and smoothing) the medians (across days) of the smallest bid–ask spreads within ±10% of at-the-money and for each of the expiration groups 0–3 months, 3–12 months, and 12–36 months are 17, 32, and 47 bps of implied volatility, respectively.

[14]  Cheridito et al. (2007) show that for square-root processes measure changes that change both the long-term level and the speed of mean-reversion are allowed.

**Table 3.** The left part of the table shows average calibration errors in basis points. Standard deviations are shown in parentheses. The right part of the table shows the average of the per-day differences in calibration errors between each pair of models. I.e., the column "Heston vs. SABR" shows statistics on the time series constructed, where on each day we subtract the Heston error from the SABR (stochastic-alpha-beta-rho) error.

| Expiry | Heston | SABR | Mixture | | Heston vs. SABR | Heston vs. Mixture | SABR vs. Mixture |
|---|---|---|---|---|---|---|---|
| 1 month | 28.7 | 11.2 | 18.4 | | 17.4 | 10.3 | −7.1 |
| | (14.8) | (9.9) | (10.3) | | (15.2) | (7.3) | (8.4) |
| 3 months | 22.7 | 9.1 | 13.1 | | 13.6 | 9.5 | −4.1 |
| | (10.1) | (7.7) | (5.7) | | (14.6) | (7.3) | (8.5) |
| 6 months | 14.8 | 11.8 | 9.9 | | 2.9 | 4.8 | 1.9 |
| | (8.9) | (8.9) | (6.0) | | (14.4) | (8.0) | (7.6) |
| 1 year | 9.5 | 17.7 | 11.2 | | −8.2 | −1.7 | 6.4 |
| | (7.3) | (9.9) | (8.7) | | (8.5) | (6.0) | (4.4) |
| 1.5 years | 10.3 | 20.9 | 14.7 | | −10.6 | −4.5 | 6.2 |
| | (9.2) | (10.7) | (10.0) | | (7.1) | (4.3) | (3.5) |
| 2.5 years | 14.2 | 23.6 | 18.4 | | −9.4 | −4.3 | 5.2 |
| | (10.0) | (12.1) | (10.9) | | (7.5) | (4.4) | (3.5) |

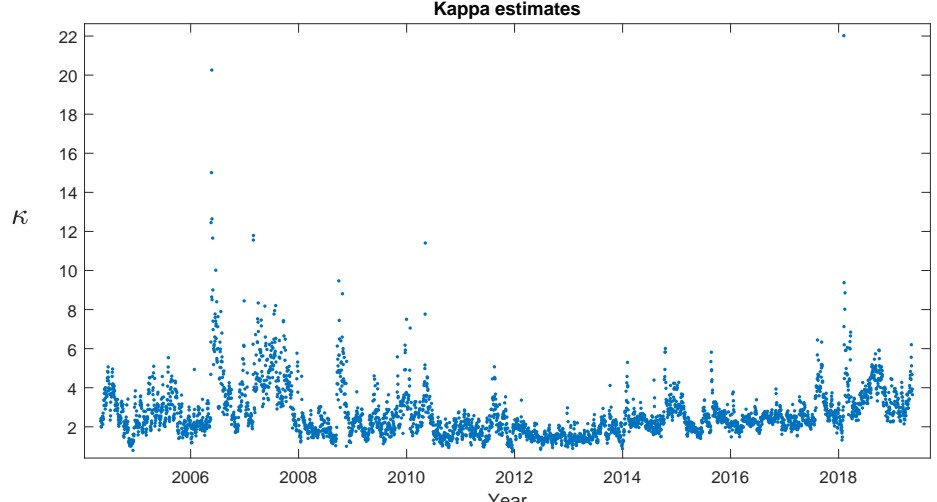

**Figure 2.** Estimates of the Heston model's $\kappa$ under the risk-neutral measure $\mathbb{Q}$ obtained from the two-step calibration procedure.

Average errors do not tell the whole story; the variation (between days) of calibration errors (as measured by their standard deviations) are of the same order of magnitude as the averages. In the right-hand part of Table 1, we give paired comparisons of the models. On each day (and for all three combinations) we subtract one model's calibration error from an other model's and report averages (that could be calculated from the left-hand part of the table) and standard deviations (that cannot). Let us first note that this shows that the differences in average that we comment on are all highly statistically significant (thus we have not cluttered the table with indications of this). However, we also note that the differences display a variation that is quite similar to the individual calibration errors themselves. Hence when the average Heston error is 14.2 bps for 2.5-year options and it is 23.6 bps for SABR, it by no means implies that Heston calibrates around 10 bps better each and every day, as the standard deviation of the difference is 7.5 bps.

As a final in-sample investigation, let us look at the time evolution of the calibration errors and the model parameters. In Figures 3–5 we therefore show such results for, respectively, 1 month, 6 months, and 2.5 years expiries. The calendar and expiry time variation that one would expect from the results in Table 3 is evident. Around the 2008–2009 financial crisis we see deterioration in model performance in various guises; Heston calibration errors for 1 month options more than

double, and the model never really recovers; the crisis leads to extreme (negative) correlation for SABR. But overall, nothing off-the-scale happened. One thing that stands out visually is how high the Heston's variance-of-variance parameter $\eta$ (left-hand side columns, 2nd panel) correlates with the level of volatility (left-hand columns, 1st panel); the average (across expires) correlation is 0.53. This is consistent with our previous elasticity estimation: The Heston model's elasticity of $\frac{1}{2}$ does not allow the variance of variance to react as strongly to changes in the variance level as empirically observed in time-series ($\lambda_v$ around 0.91–0.97), and that manifests itself as changes in the $\eta$-estimate. For SABR, the (volatility, vol-of-vol) correlation is mildly negative ($-0.28$ on average), which is also consistent with that model slightly overstating the elasticity. A final note to make (based on the top left panel in Figure 3) is that even though implied, at-the-money volatility goes to instantaneous volatility $\sigma_t$ as time to expiry goes to 0, and a one month expiry is not sufficiently small for this asymptotic result to have kicked in.

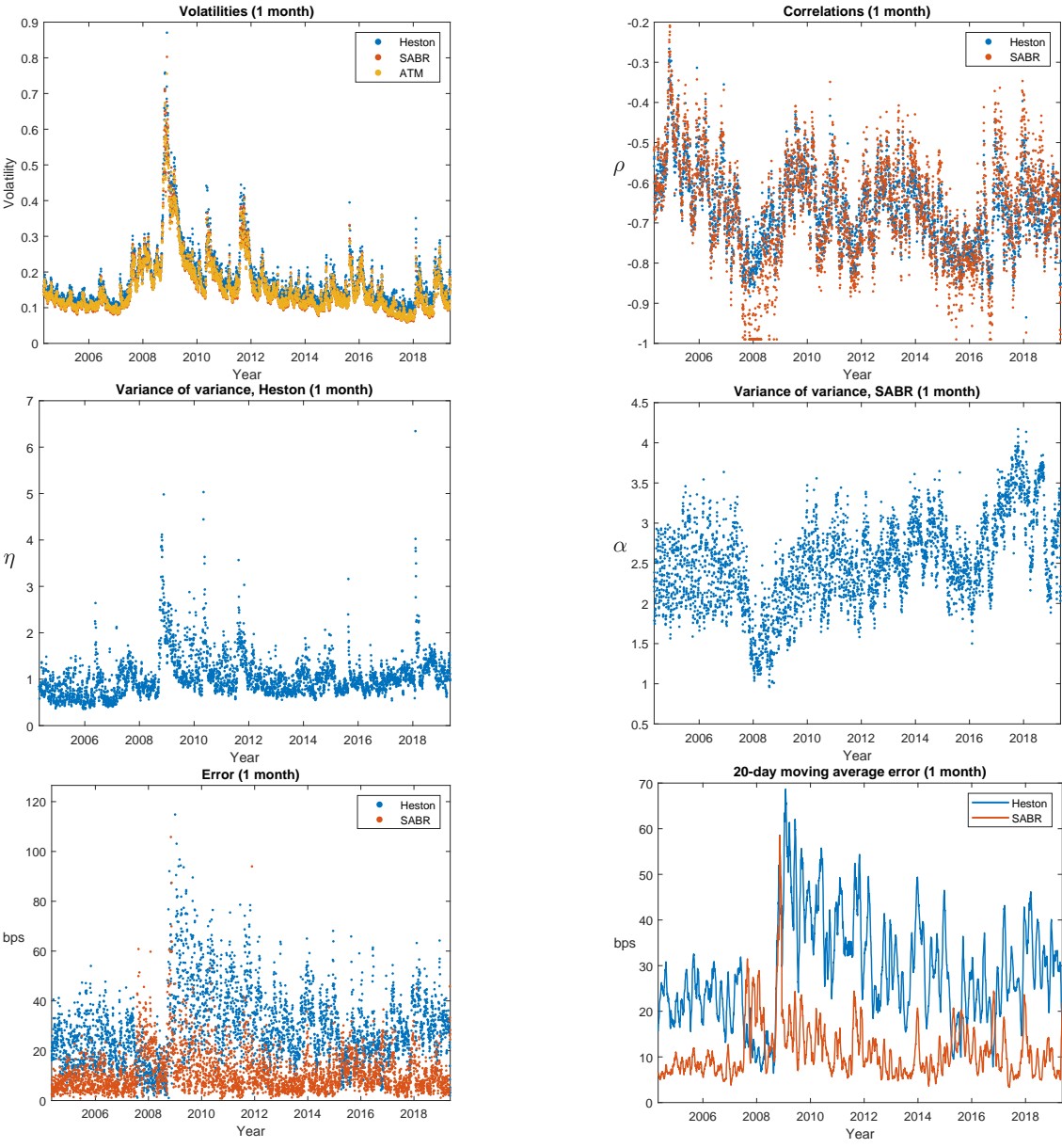

**Figure 3.** Calibration to the 1 month expiry. To improve visibility, we have in the bottom left plot excluded the extreme errors on the 21st of November 2008 of 260 bps (Heston) and 352 bps (SABR).

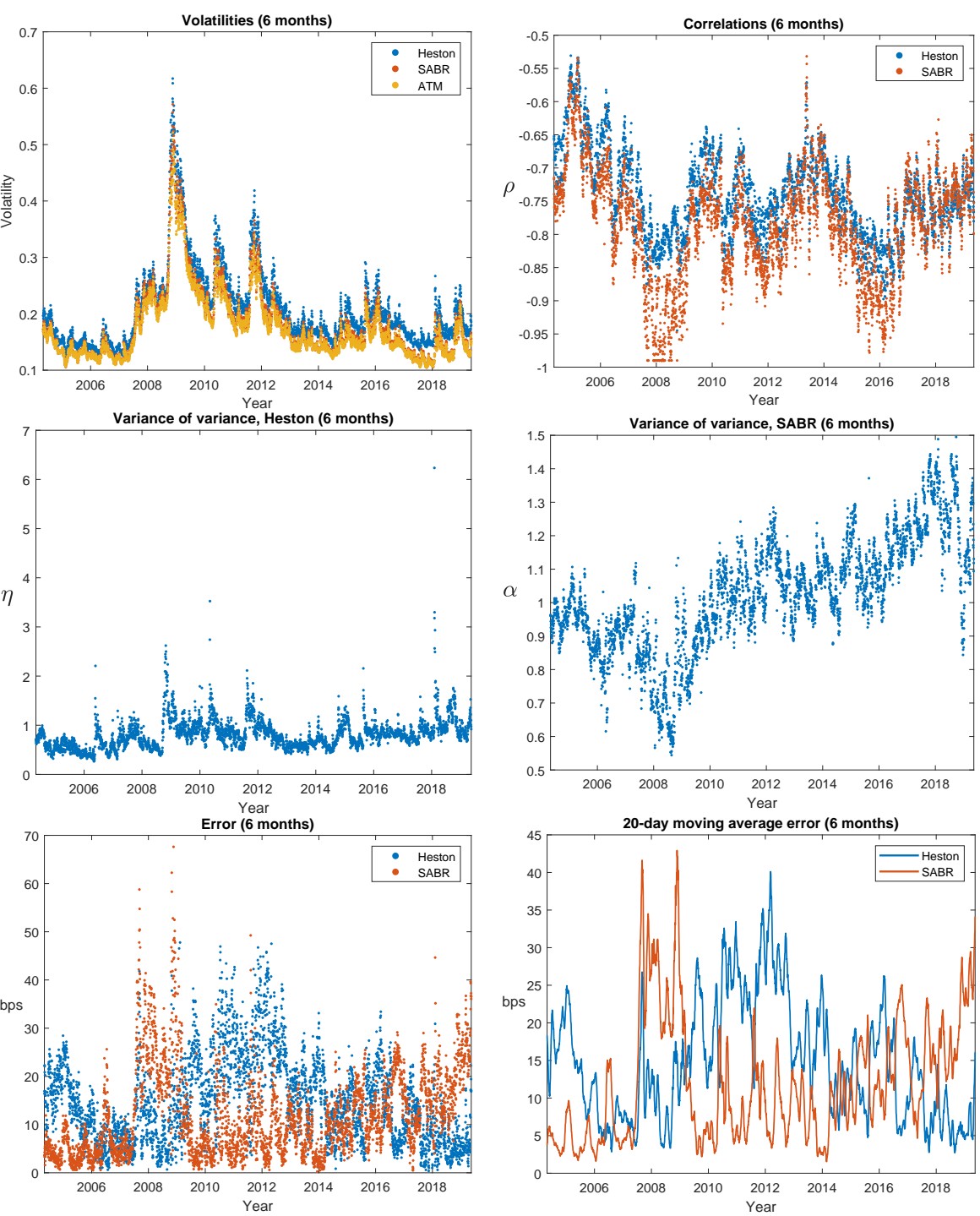

**Figure 4.** Calibration to the 6 month expiry.

### 3.2. Predictions and Hedging; Out-of-Sample Performance

Figures 3–5 show that calibrated model parameters change both over calendar time and across expiries. A cynic would say, "Therefore they are not *parameters*—which is a crucial assumption in analytical work with the models, e.g., derivation of option price formulas. So: back to the drawing-board". Our defense against that argument is pragmatism as formulated in the famous quote from statistician George Box that "all models are wrong, some are useful". Thus we now investigate the practical usefulness of the SABR and Heston models. More specifically, we look at the quality of predictions and at how helpful they are for constructing hedge portfolios; two aspects that

are central to financial risk management. Both investigations are done in out-of-sample fashion; the predictions or portfolios made at time $t$ use only information that is available at time $t$.

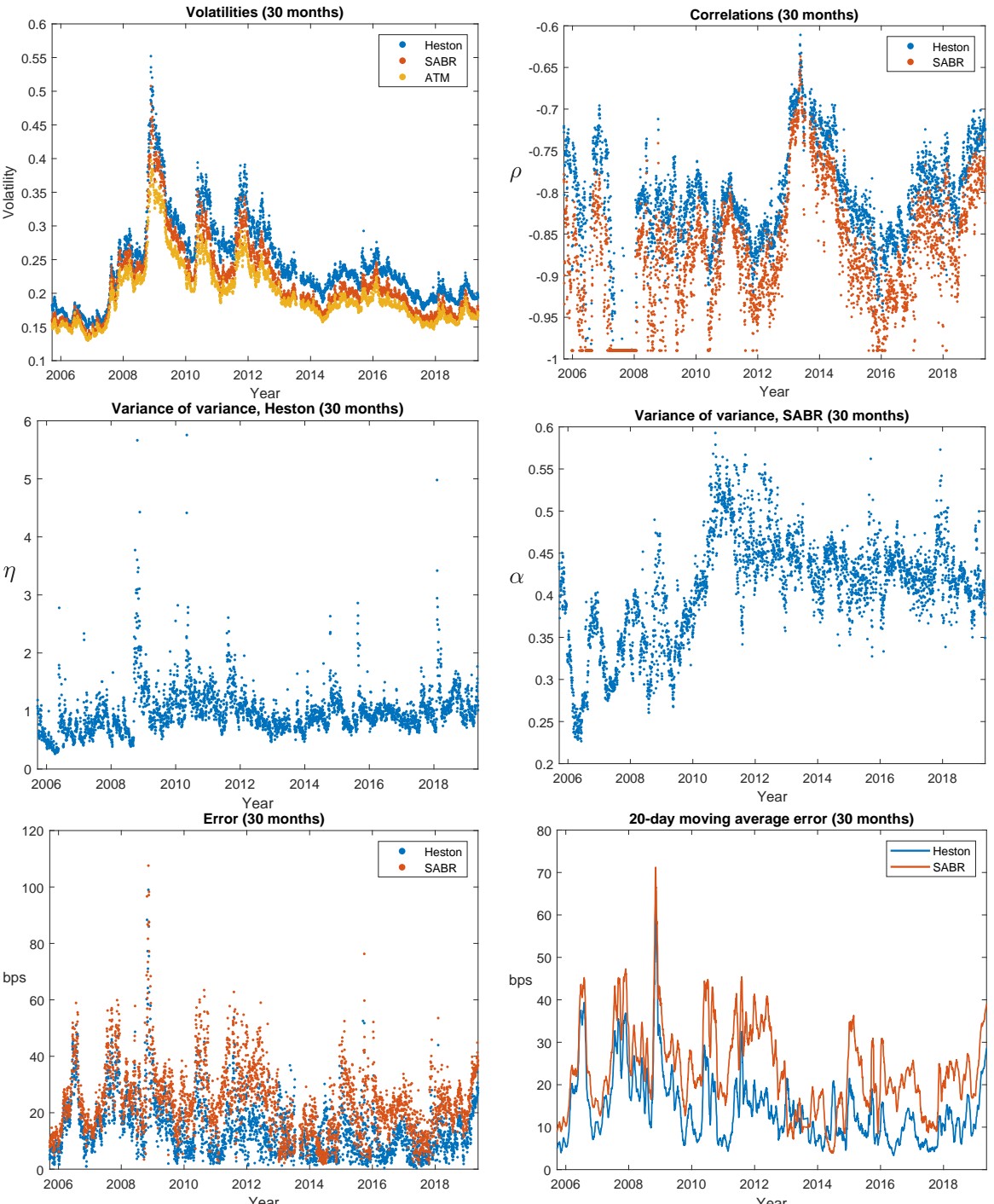

**Figure 5.** Calibration to the 30 months (2.5 years) expiry.

To study prediction quality, we conducted the following experiment: on each trading day we calibrated the parameters and volatility. We then moved forward in time, updated observable market variables, such as the index price as well as yields, and recalibrated the volatility (which is allowed to change in the model)—but *not* the parameters. We performed the experiment by moving 1 to 20 trading days ahead (the horizon) and considering all possible starting dates. Figure 6 shows the results, and more specifically average absolute errors at different horizons and for different expiries for the

SABR and Heston models.[15] The results are consistent with what we have observed so far. SABR works better for short expiries (where errors are generally larger), while Heston does better for longer expiries. Average absolute errors are well fitted by square-root functions (which is how we would a priori expect standard deviation-like quantities to grow with time), and the difference between the models is stable across horizons.

Finally, we turn to the hedge performance of the models. We will attempt to hedge out-of-the-money options by trading appropriately in the underlying asset, the risk-free asset as well as an at-the-money option of the same type as the out-of-the-money option (i.e., call or put). We performed the analysis on out-of-the-money options with strikes corresponding to Black–Scholes Delta values of 0.1, 0.3, 0.7, and 0.9. Thus, if the Delta is below 0.5 we hedged a call option and otherwise hedged a put option.

The details of the experiment are as follows: on each day we sell the out-of-the-money option and form a portfolio that according to the calibrated model perfectly hedges this option. Let us write $h_t := (h_t^b, h_t^s, h_t^a, h_t^o)$ to denote the entire portfolio time $t$, where $h_t^b$ denotes the number of units in the risk-free asset, $h_t^s$ the number of units in the underlying asset, $h_t^a$ the number of units in the at-the-money option, and $h_t^o$ the number of units in the out-of-the-money option. We set $h_t^o = -1$ as mentioned. Let us also write $\pi_t^o$ and $\pi_t^a$ to denote the observed values of the out- and at-the-money option, respectively.

For compactness of presentation, let us consider a general hedge model

$$dS_t = S_t(r - q)dt + S_t\sqrt{V_t}dW_{1,t} \tag{19}$$

$$dV_t = a(t, S_t, V_t)dt + b(t, S_t, V_t)\eta\sqrt{V_t}dW_{2,t} \tag{20}$$

where $\langle dW_{1,t}, dW_{2,t} \rangle = \rho dt$ and where $a$ and $b$ are functions. We get Heston with $a(t, s, v) = \kappa(\bar{v} - v)$, $b(t, s, v) = 1$, and SABR with $a(t, s, v) = \alpha^2 v$, $b(t, s, v) = \sqrt{v}$ and $\eta = 2\alpha$. We can write the price of the out-of-the-money option at time $t$ as $F(t, S_t, V_t; \theta)$ for an appropriate function $F$ and a parameter vector $\theta$ depending on model specifics; $\theta := (\eta, \rho, \bar{v}, r, q)$ for Heston and $\theta := (\alpha, \rho, r, q)$ for SABR. Similarly, the price of the at-the-money option can be written as $G(t, S_t, V_t; \theta)$ for an appropriate function $G$. To determine the perfectly replicating portfolio under the hedge model we start by computing our models *Delta* and *Vega*, which for the out-of-the-money option with function $F(t, s, v; \theta)$ will be defined as $F^s := \frac{\partial F}{\partial s}$ and $F^v := \frac{\partial F}{\partial v}$, respectively.[16] Our hedge will now consist of

$$h_t^a = \frac{F_t^v}{G_t^v} \tag{21}$$

in the at-the-money option, and

$$h_t^s = F_t^s - h_t^a \cdot G_t^s \tag{22}$$

in the underlying asset. The joint portfolio will then be kept self-financing with the risk-free asset. With these choices the associated value process will exactly be a function of the current state $(t, S_t, V_t)$. Computing the dynamics of it using Ito's Formula and applying the principle of no arbitrage proves that $h_t$ is in fact a perfect hedge of the out-of-the-money option—assuming we are using the correct model.

---

[15] Since the mixture model is underspecified and therefore does not imply a specific dynamic structure, we cannot move the model forward in time without further assumptions. We therefore excluded it from this experiment.

[16] As done here, we will often suppress the input arguments to simplify the notation. We will also write $F_t^s := \frac{\partial}{\partial s}F(t, S_t, V_t; \theta)$, etc. when we need to stress the time point used. We will use similar notation to denote other derivatives of $F$ and do all of this also for the function $G(t, s, v; \theta)$.

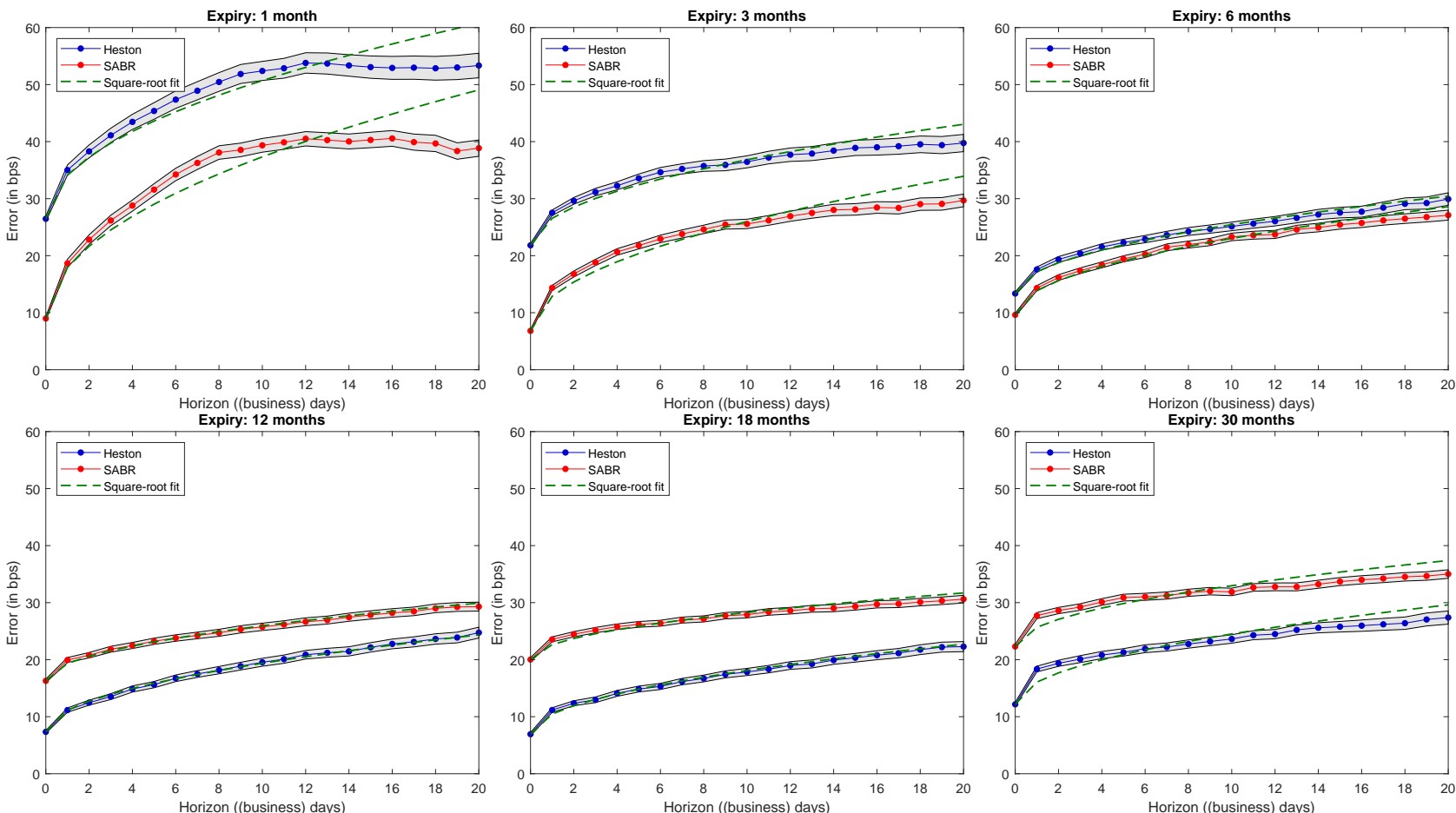

**Figure 6.** Model prediction quality. The graphs show average absolute errors at different horizons (x-axis) and for different expiries (panels). Blue is Heston and Red is SABR. The shaded areas are 95% confidence bands, the dotted green curves are the best-fitting $\sqrt{x}$-functions.

For the mixture model the whole problem of finding a perfectly replicating portfolio is ill-defined, since the dynamical structure is unspecified. We therefore instead test a mixed *portfolio* that is the average of the portfolios under Heston and SABR. While this portfolio may not perfectly replicate the option it will be a valid self-financing strategy, which we can compare to the pure Heston and SABR strategies.

Consider now discrete hedging between two trading days $t_i$ and $t_{i+1}$ with $\Delta t = t_{i+1} - t_i = 1/252$ assumed for simplicity. Letting $V_t^h$ denote the actual value of the portfolio at time $t$ we record (with a few discrete approximations) the change in the value process from $t_i$ to $t_{i+1}$ as

$$\Delta V_{t_i}^h = h_{t_i}^b B_{t_i} r_{t_i} \Delta t + h_{t_i}^s \Delta S_{t_i} + h_{t_i}^a \Delta \pi_{t_i}^a + h_{t_i}^o \Delta \pi_{t_i}^o + h_{t_i}^s S_{t_i} q_{t_i} \Delta t. \tag{23}$$

where $B_t$ is the value of the risk-free asset. We then define the (relative and discounted) hedge error as

$$\begin{matrix} \text{Hedge error} \\ \text{(relative and discounted)} \end{matrix} = 100 \times \frac{e^{-r_{t_i}\Delta t} \Delta V_{t_i}^h}{\pi_{t_i}^o}. \tag{24}$$

and summarize the performance of each model by taking the standard deviation of this across all trading days.

In Table 4 we show the results across expires, moneyness, and models. We first note that the more out-of-the-money the target option is, the more difficult it is to hedge; not surprising, particularly because the at-the-money option is one of the hedge instruments. Short-expiry options are more difficult to hedge; above 6 months expiry standard deviations are quite stable. For low-strike options (i.e., out-of-the-money puts) the differences between models are small: at most 0.5 percentage (the 1 month, lowest-strike case), and none of the differences are statistically significant at the 5%-level. But there is an asymmetry. For high strikes (i.e., for out-of-the-money calls) differences in hedge errors are statistically significant across models; SABR outperforms Heston for expiries of three months and below, but is beaten by Heston for expiries above one year, albeit with a lower absolute margin. We also see that except in a single, statistically insignificant case (highest strike, 6 months expiry) the mixture strategy is dominated in terms of hedge performance by either Heston or SABR, i.e., there are no benefits from model "averaging" or "diversification". Results (not reported) are qualitatively similar in a hedge experiment where Vega-hedging is done only weekly.

**Table 4.** Hedge errors (in the scaled standard deviation sense of Equation (24)) from daily hedging for each expiry, model and moneyness (measured by the Black–Scholes call Delta $\Delta_{BS}$). The symbols ‡ and † indicate significance at the 1% and 5% levels for testing if the standard deviation of Heston and the mixed portfolio, respectively, are different from SABR. The symbols therefore only appear for rows related to Heston and the mixed portfolio. Significance is tested using the Brown–Forsythe test of Brown and Forsythe (1974) with the central locations estimated by the medians.

| | | \multicolumn{4}{c}{Standard Deviation of Hedge Error (Daily Hedging)} | | | |
| Expiry | Model | $\Delta_{BS} = 0.9$ | $\Delta_{BS} = 0.7$ | $\Delta_{BS} = 0.3$ | $\Delta_{BS} = 0.1$ |
|---|---|---|---|---|---|
| 1 month | Heston | 9.8 | 3.2 | 6.2 ‡ | 16.6 ‡ |
| | SABR | 9.3 | 3.2 | 4.3 | 14.7 |
| | Mixed portfolio | 9.3 | 3.2 | 5.1 ‡ | 15.2 |
| 3 months | Heston | 5.3 | 1.4 | 2.5 ‡ | 9.5 ‡ |
| | SABR | 5.2 | 1.4 | 1.8 | 8.8 |
| | Mixed portfolio | 5.2 | 1.4 | 2.1 ‡ | 8.9 |
| 6 months | Heston | 3.7 | 1.0 | 1.4 ‡ | 5.7 |
| | SABR | 3.7 | 1.0 | 1.2 | 5.5 |
| | Mixed portfolio | 3.7 | 1.0 | 1.3 | 5.4 |
| 1 year | Heston | 3.1 | 0.8 | 1.0 | 4.5 ‡ |
| | SABR | 3.1 | 0.8 | 1.0 | 4.8 |
| | Mixed portfolio | 3.1 | 0.8 | 1.0 | 4.6 † |
| 1.5 years | Heston | 2.5 | 0.6 | 0.9 ‡ | 3.5 ‡ |
| | SABR | 2.5 | 0.7 | 1.0 | 3.9 |
| | Mixed portfolio | 2.5 | 0.7 | 0.9 | 3.6 ‡ |
| 2.5 years | Heston | 3.7 | 0.8 | 1.1 | 4.4 ‡ |
| | SABR | 3.8 | 0.8 | 1.2 | 4.7 |
| | Mixed portfolio | 3.7 | 0.8 | 1.1 | 4.5 |

## 4. Conclusions

The one-word answer to the question in the title would be "lognormal-ish". We find that the dynamics of the instantaneous variance of the S&P 500 index is best described by a model with elasticity slightly below one, i.e., a model with a close-to lognormal volatility structure. However, one-factor diffusion models fail to capture the empirical auto-correlation structure of instantaneous volatility at horizons above a few weeks; long-memory even in the rather short run, as it were. For option pricing, the lognormal SABR model performs best for short-expiry options (expiries of 1–3 months) and the Heston model performs best for expiries of one year and longer. We ascribe the latter effect to the Heston model's mean-reversion—which one should treat with care as its force is quite different under $\mathbb{P}$ and $\mathbb{Q}$. A simple mixture of the two models does not yield benefits. These results are robust across time (15 years of option data) and both in-sample and out-of-sample, including predictions and hedging.

**Author Contributions:** Conceptualization and writing, S.E.R. and R.P.; software and data curation, S.E.R. Both authors have read and agreed to the submitted version of the manuscript. All authors have read and agreed to the published version of the manuscript.

**Funding:** This research received no external funding.

**Conflicts of Interest:** The authors declare no conflict of interest.

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
