# Peer review of "How Does the Volatility of Volatility Depend on Volatility?"

_risks, doi:10.3390/risks8020059_

Round 1

Reviewer 1 Report

The avthor found that the dynamics of the instantaneous variance of the S&P 500 index is best described by a model with elasticity slightly below one, i.e. a model with a close-to lognormal volatility structure. For option pricing, the lognormal SABR model performs best for short-expiry options (expiries 1-3 months) and the Heston model performs best for expiries of one year and longer.

Paper is well written and is suitable for publication in this journal.

Author Response

We are very happy that Reviewer 1 liked the paper.

In the revised version (where all substatial changes are written in red) we have (primarily at the end of section 2) extended the analysis of some of the empirical problems of modelling volalitility by a one-factor diffusion, long-memory in particular.

Reviewer 2 Report

Report on “How Does the Volatility of Volatility Depend on Volatility?” by

Sigurd Emil Rømer, Rolf Poulsen

The authors study the  variance of variance  parameter that comes out of the SABR model, the Heston model and a hybrid average of the two. They conclude that the
SABR model performs best for short-term options, the Heston model is superior for long-term options, and a mixture of the two models does not lead to improvements.

I find that their analysis appears sound, but I have some deeper concerns regarding properties of the Heston and SABR models (even though they are very helpful,  popular and widely used).

I would prefer to see the following points addressed to some degree in their paper:

    1. One should look at the distribution of  returns and other stylized facts  implied by the models, and compare these to those of S&P 500 returns. Due to the convolution properties of the underlying models it is very likely that neither produces the stylized facts of actual returns over timescales must larger that one time step. Basically, the Heston model convolves way too quickly to a Gaussian distribution and the fat tails of real returns which exist on time scales up to 2 weeks to  a month are not captured. In the  SABR model, one can tune the parameters to capture the fat tails, but  for a constant parameter the observed power law distribution over multiple time horizons is not captured. This aspect should definitely be looked at and discussed; if the underlying models cannot accurately describe the real returns distribution then how can they adequately model elasticity of variance, or fit options data over all horizons?

2. Memory: memory in volatility is well documented, a close-to  log-normal distribution of volatility is also well documented  in the literature.  There are papers that propose models  that capture realistic  behavior of the underlying volatility including the memory, and the  conditional volatility which is related to the volatility of the volatility.   It might well be that the reason that the authors don’t find that either SABR or Heston yield really good fits to both short- and long-term expirations could be due to the fact that the memory is not not captured accurately in either of these models. They do elude to such a possibility  in the paper: “all models are wrong, some are useful”, but perhaps they could look at, and highlight, this particular short-coming in the models under study. Volatility models that DO capture memory are  FSV models, GARCH models, multi time-scale models, and multi fractal models, and although they don’t offer closed form option pricing formulas. But if the topic under study is “the volatility dependent volatility of volatility” then these models might be better candidates.

Some long range memory models:

  •     Gatheral, Jim, Thibault Jaisson, and Mathieu Rosenbaum. 2018. “Volatility is rough” Quantitative Finance 18:33-949.
  • G. Zumbach, “Volatility processes and volatility forecast with long memory”, Quantitative Finance 4 70 (2004),
  • “On a multi-timescale statistical feedback model for volatility fluctuations”Journal of Investment Strategies. ISSN: 2047-1238 (print). 2047-1246

To summarize:

           I think the paper is fine in terms of analyzing the volatility dependent volatility of volatility based on the Heston and SABR models, and their techniques appear sound and interesting. However, I think they should include and emphasize  the larger context in particular the fact that volatility has long-range memory which is not incorporated in the models they studied. Ideally it would be great  if they could study some of those models as well (at least in the section on realized variance).

Author Response

We thank the reviewer for very insightful comments, which we believe have improved to paper significantly.

All substantial changes in the revised version are written in red.

The revised version includes (lines 105-157) an analysis of some of the failings of one-factor models (and proposed solutions -- briefly), in particular the long-memory problem you point out that manifests itself in a clear difference between the model's cumulative autocorrelations and their empirical conterparts, see Figure 1. These observations have led to some addtional remarks in the abstract, introduction and conclusion.

Reviewer 3 Report

Please, see the attached pdf file

Author Response

We thank the reviewer for insightful comments that we believe have led to considerable improvements of the manuscript in its resubmitted version. In this version all (major) changes have been written in red.

Detalied responses to the points rasied in the referee report are given (as "sticky notes") in the attached pdf. 

Round 2

Reviewer 2 Report

The paper is much more complete now. Thank you for including the additional discussions.